# Eukaryotic Translation Elongation Factor *OsEF1A* Positively Regulates Drought Tolerance and Yield in Rice

**DOI:** 10.3390/plants12142593

**Published:** 2023-07-08

**Authors:** Qing Gu, Junfang Kang, Shuang Gao, Yarui Zhao, Huan Yi, Xiaojun Zha

**Affiliations:** College of Life Sciences, Zhejiang Normal University, 688 Yingbin Road, Jinhua 321004, China

**Keywords:** rice, *OsEF1A*, drought stress, yield

## Abstract

Drought is one of the most serious stresses affecting rice growth. Drought stress causes accelerated senescence, reduced fertility, and subsequent reductions in crop yield. Eukaryotic translation elongation factor EF1A is an important multifunctional protein that plays an essential role in the translation of eukaryotic proteins. In this study, we localized and cloned the *OsEF1A* gene in rice (*Oryza sativa*) in order to clarify its role in drought tolerance and yield. Subcellular localization revealed that it was mainly localized to the cell membrane, cytoskeleton and nucleus. Compared with the wild-type, *OsEF1A* overexpressing transgenic plants had significantly more tillers and grains per plant, resulting in a significantly higher yield. Increases in the relative water content and proline content were also observed in the transgenic seedlings under drought stress, with a decrease in the malondialdehyde content, all of which are representative of drought tolerance. Taken together, these findings suggest that *OsEF1A* plays a positive regulatory role in rice nutritional development under drought stress. These findings will help support future studies aimed at improving yield and stress tolerance in rice at the molecular level, paving the way for a new green revolution.

## 1. Introduction

Rice (*Oryza sativa*) is one of the most important food crops in China; however, abiotic stresses such as drought, heat, flooding and salinity all have a significant effect on crop yield. China’s arid and semi-arid arable land area accounts for about 1/2 of the national land area, and there is a growing trend of expansion. Drought is one of the most frequent, long-lasting and serious natural disasters affecting rice yield, as characterized by decreases in the number of spikelets per panicle, the number of grains per plant and the 1000-grain weight. In China, the increasing population and reductions in arable land are resulting in urgent food shortages and an increased demand for rice, highlighting the need for cultivars possessing increased drought tolerance and improvements in yield [1,2,3].

Drought stress causes accelerated senescence, reduced fertility, and subsequent reductions in crop yield. Accordingly, a number of studies in rice have examined the selection of drought-tolerant varieties [4,5], agronomic cultivation measures [6,7], cellular signaling response mechanisms [8], and stress physiology and related genes [9,10,11] with the aim of understanding the intrinsic mechanisms of drought tolerance. For example, Zha et al. [12] found that overexpression of *LRK1* in the LRK gene cluster increased the number of tillers and grains per plant in rice, thereby resulting in an increase in yield. Meanwhile, Kang et al. [13] found that the overexpression of *LRK2* increased the effective tiller number, crop yield and enhanced tolerance to certain abiotic stresses. Using a yeast two-hybrid assay and bimolecular fluorescence complementation (BIFC), they also revealed an interaction between the *LRK2* gene and elongation factor *OsEF1A* in vitro.

Two types of elongation factors (eEF) exist in plants, tetrameric eEF1 and monomeric eEF2 [14]. The eukaryotic translation elongation factor *OsEF1A*, a multifunctional protein within the G protein family of translation eEF, plays an important role in the translation of eukaryotic proteins [15]. The *OsEF1A* complex is formed via a subunit of eEF1 with high affinity for guanosine triphosphate (GTP), mediating the binding of aminyl tRNA to ribosomes [16]. Recent studies have shown that *OsEF1A* also interacts with a number of proteins involved in plant virus replication and propagation [17,18], mediation of cell death [19,20] and early leaf senescence [21], as well as playing a role in stress resistance [22]. For example, a positive correlation between eEF1A and the protein lysine content was observed in the maize endosperm [23,24], while a role in heat stress was observed in potatoes [25]. Since 1988, a total of four different elongation factors encoding protein synthesis (EF1A) have been cloned in rice [26]; however, few studies have been carried out.

A total of 14 genes encoding EF1A-like proteins have been reported in the rice genome, of which four encode EF1A. Previous studies in indica rice revealed a significant increase in *OsEF1A* expression under abiotic stress [27]. In this study, we therefore cloned and characterized *OsEF1A*, and examined transgenic *OsEF1A*-overexpressing lines, revealing an increase in effective tiller number, yield and drought tolerance compared to wild-type rice. *OsEF1A::GUS* was expressed mainly in the roots, node, basal stem and anther. Meanwhile, under drought stress, the relative water content, chlorophyll content and proline content increased, while the malondialdehyde content significantly decreased in the *OsEF1A*-overexpressing plants, all of which led to drought tolerance.

## 2. Results

### 2.1. Confirmation of OsEF1A as a Translation Elongation Factor

In rice, *OsEF1A* encodes 447 amino acids and is thought to consist of three structural domains: a GTP binding domain, domain 2 and domain 3 (Figure 1A). Phylogenetic comparison of EF1A (GenBank accession number: GQ848073.1) between different species revealed the following sequences: DQ174254.1, KU886197.1, NC 038255.2, NC 003070.9, AT1G07920.1, DV858943.1, AF479046.1, EZ421973.1, NC 003076.8 (Figure 1B). Based on these findings, *OsEF1A* was found to be most closely related to CT841797.1 in wild rice in China (*Oryza rufipogon*).

### 2.2. Characteristics of OsEF1A in Rice

To determine the subcellular localization of *OsEF1A*, the *OsEF1A* gene was fused to a CFP gene expression vector using the cauliflower mosaic virus 35S promoter (CaMV35S) then introduced into onion epidermal cells using the gene gun method. Compared to the control, the *35S::CFP:OsEF1A* construct was mainly localized to the cell membrane, cytoskeleton and nucleus under laser confocal microscopy (Figure 2A). To further examine the expression pattern of *OsEF1A*, an *OsEF1A::GUS* expression vector was constructed and then transferred into wild-type rice using Agrobacterium rhizogene-mediated gene transformation. Tissue samples were then obtained at the flowering stage for GUS staining and analysis of expression patterns, as shown in Figure 2B. Expression was mainly observed in the roots, node, basal stem and anther. In contrast, no expression was observed in the control wild-type plants.

To determine whether *OsEF1A* is affected by drought stress, two-week-old wild-type rice seedlings were subjected to 20% PEG6000 treatment then total RNA was extracted at 0, 3, 6, 12 and 24 h after treatment and then quantified using real-time PCR. The results showed that *OsEF1A* expression was induced under drought stress, with an increase of approximately 7.4-fold at 12 h after treatment. Then, it dropped to about 3.3-fold at 24 h (Figure 2C). These findings suggest that increased expression of *OsEF1A* may be an adaptive response to environmental stress.

### 2.3. Plasmid Construction and Rice Transformation

To further understand the role of *OsEF1A*, *2 × 35S::OsEF1A* and *2 × 35S::antiOsEF1A* plasmids were constructed (Figure 3A). T0 generation plants overexpressing *OsEF1A* and antisense *OsEF1A* lines were also obtained by transferring an expression vector containing the full *OsEF1A* gene into the wild-type using Agrobacterium-mediated transgenic technology. DNA was subsequently extracted from the leaves of seedlings then *pCAMBIA1300s* vector Hygromycin B phosphotransferase gene-specific primers were used to identify transformation success using *Oryza sativa* L. spp. *Japonica* as a negative control (Figure 3B). Amplification of an electrophoretic band approximately 400 bp in size was observed in most seedlings compared to the wild-type. Four T2 generation transgenic lines and the wild-type were selected for expression analysis at the three-leaf stage using real-time PCR. Accordingly, *OsEF1A* expression was stronger in the OE-1 and OE-4 lines and reduced in AOE-6 and AOE-8 (Figure 3C).

### 2.4. Overexpression of OsEF1A Improves Rice Yield

To investigate the effect of *OsEF1A* on yield traits, *OsEF1A*-overexpressing transgenic lines were cultivated under the same conditions as the wild-type (20 individual plants per line). The following yield components were then examined: number of tillers, plant height, 1000-grain weight, seed setting rate and the number of grains per plant (Table 1). At the tillering stage, overexpression of *OsEF1A* resulted in more significant tillers development (Figure 4A), with a significant increase in the effective tiller number of 28.74% and a decrease in plant height of 8.60% compared to the wild-type. Meanwhile, at the mature stage (Figure 4B), the overexpressing plants showed a significant increase in the number of grains per plant (Figure 4C) but a decrease in both the 1000-grain weight and seed setting rate. The number of grains per plant is affected by the number of primary and secondary branching, which were also counted in the overexpressing and wild-type plants. Accordingly, an increase in both primary and secondary branching was observed in the transgenic compared to wild-type plants (Table 1, Figure 4D). These results indicate that the *OsEF1A* gene positively regulates tillering and yield in rice.

### 2.5. Overexpression of OsEF1A Improves Drought Resistance in Rice

The phenotypes of wild-type rice, *OsEF1A*-overexpressing rice (OE-1 and OE-4) and the *OsEF1A*-antisense lines (AOE-6 and AOE-8) were subsequently investigated under drought stress. The germination rates of the transgenic and wild-type plants were first analyzed under drought treatment to confirm the role of *OsEF1A* in drought tolerance. After treatment for 4 days, the germination rates were significantly higher in the OE-1 and OE-4 lines than the wild-type, while those in AOE-6 and AOE-8 were lower (Figure 5A,B). The seedlings were then treated with 20% PEG 6000 for 14 days, resulting in yellowing of the leaf tips on all plants after treatment for 5 days. And after treatment for 10 days, the OE-1 and OE-14 lines showed less wilting than the wild-type, while the AOE-6 and AOE-8 lines were severely affected (Figure 5C). For all lines, the treatment ends on day 10 and culture is resumed with water. The OE-1 and OE-4 lines were significantly greater compared to the wild-type. Survival rates were 62.5 and 68.75% in the OE-1 and OE-4 lines compared to significantly lower rates of 12.5 and 14.58% in AOE-6 and AOE-8, respectively (Figure 5D). These results suggest that survival under drought stress was significantly better in the *OsEF1A*-overexpressing plants compared to the wild-type, suggesting that the overexpression of *OsEF1A* plays a positive role in regulating the drought stress response in rice.

The relative water content, malondialdehyde, proline and chlorophyll contents were subsequently measured in the wild-type and overexpressing plants after 4 d of drought stress treatment. The relative water content and proline content increased significantly in the overexpressing lines compared to wild-type plants (Figure 6A). Proline is involved in osmoregulation within the cell, osmotic balance inside and outside, and integrity of the cell membranes (Figure 6B). Meanwhile, malondialdehyde, which is an important indicator of membrane oxidative damage, decreased in the overexpressing plants compared to the wild-type, indicating an effective reduction in oxidative damage under drought stress (Figure 6C). In contrast, no significant differences in chlorophyll content were observed between the different lines (Figure 6D). Taken together, these results suggest that overexpression of *OsEF1A* improves drought resistance in rice.

## 3. Discussion

Drought is one of the most severe and widespread natural disasters in the world. Drought poses a serious threat to food and ecological security and has become one of the major constraints of sustainable development in socio-economic systems [28]. In this study, we identified the function of the translation elongation factor *OsEF1A* in rice yield and drought stress. The results showed an increase in yield in rice plants overexpressing *OsEF1A*, with significant increases in effective tiller number and grains per plant compared to the wild-type. In addition, the overexpressing plants also showed greater drought tolerance under drought stress. *OsEF1A* was mainly localized in the cell membrane, cytoskeleton and nucleus, and was expressed in the roots, node, basal stem and anthers. Under the 20% PEG6000 treated, the *OsEF1A* was increased at 12 h. These results highlight the potential of *OsEF1A* in improving rice yield in arid regions.

Earlier studies revealed that overexpression of certain genes results in an increase in crop yield or enhanced abiotic stress resistance [29,30,31,32]. For example, overexpression of *RAG2*, a rice 16-kDa amylase/trypsin inhibitor, significantly increased rice grain size and 1000-kernel weight and improved both grain quality and yield, with significant differences in major seed storage material between the *RAG2*-high-expressing and *RAG2*-inhibiting strains compared with the wild type. Protein content and total lipid content were increased and decreased in the *RAG2* high expression and *RAG2* suppressor strains, respectively [33]. Meanwhile, overexpression of *OsbHLH120* resulted in an increase in drought stress resistance and increased expression of ABA synthesis genes [34]. In addition, blocking of *Osa-miR1871* was found to improve resistance to rice blast fungus and simultaneously improve grain yield. Moreover, miR1871 was found to regulate rice yield and immunity through *MFAP1*, suggesting that the *miR8171-MFAP1* module could be used for rice breeding aimed at improved immunity and yield [35]. Meanwhile, heterologous overexpression of *OsbZIP5* in Arabidopsis reduced root length and drought tolerance under polyethylene glycol and abscisic acid treatments, while down-regulation of *OsbZIP5* gene expression improved physiological indicators (such as proline, malondialdehyde and chlorophyll) and reduced stomatal opening and water loss in transgenic rice, acting as a negative regulator under drought stress in rice [36].

Branches occur below or near the ground in plants such as grasses. It arises from the more expanded and nutrient-rich tiller nodes. The tiller that emanates directly from the tiller node at the base of the main stem is called primary tiller, and new tiller buds and adventitious roots can be produced at the base of primary tiller to form secondary tiller. Under good conditions, the third and fourth level tillers can be formed. As a result, a plant forms with many branches clumped together. Tillering is an important factor for yield, and accordingly, several genes associated with tillering have been identified [37,38,39]. For example, *OsWUS* [40] was found to promote growth of rice tillering shoots by affecting apical dominance; Rice euAP2-like transcription factor *OsAP2-4* may be involved in the regulation of axillary bud differentiation and lateral branch elongation through the regulation of the solanum lactone signaling pathway, which affects the regulation of rice tillering and plant establishment [41]; while *LAZY1* [42] and *LAZY2* [43] were found to regulate the angle of tillering. In addition, in recent years, it has also been found that plant hormones also have important effects on rice tillering, such as growth hormone, cytokinin and diclofenac lactone. Growth hormone and diclofenac lactone inhibit the growth and development of tillering buds, while cytokinin promotes the growth of tillering buds, and plant genes such as oleuropein lactone and jasmonic acid also affect rice tillering to some extent [44,45,46]. In this study, The transgenic strain overexpressing *OsEF1A*, a downstream protein that interacts with the *LRK* gene, showed a significant increase in the effective number of tillers and grains per plant, and the GUS staining results showed that *OsEF1A* was expressed at the root-stem union and at the base of each stem node indicating that *OsEF1A* has an extremely important effect on the gene for tillering during plant growth and development, consistent with the fact that the overexpression of *LRK1* and *LRK2* both increased the effective number of tillers in rice, and *LRK1* also increased the number of grains per plant, but *LRK2* did not increase the affiliation per plant very significantly, indicating that the overexpression of *OsEF1A* has a regulatory role in improving rice yield. However, the agronomic trait statistics revealed that the plant height, thousand grain weight and fruiting rate of the overexpression strain were significantly lower than those of the wild type, which might be related to the increase in tiller number, while the uptake of nutrients in the low medium by the roots was fixed, leading to a decrease in organic matter synthesis and a significant decrease in stem node length and fruiting rate. The specific mechanism of the *OsEF1A* gene on rice tiller number improvement is still unclear and more in-depth studies are needed.

During the growth of rice, a large amount of water is required. Drought stress decreases the water potential in the soil, resulting in the inability of cells to maintain expansion pressure, accompanied by an increase in the content of reactive oxygen species in the cells, which causes oxidative damage to biomolecules such as membrane lipids and proteins, leading to yield reduction or death of rice. When rice resists drought stress, it will cause a series of physiological and biochemical reactions in the rice plant, such as the synthesis of some antioxidant enzymes (SOD, POD, CTA) to remove excessive reactive oxygen species from the cells or accumulate a certain level of soluble sugars and proline in the cells to maintain osmoregulation, retain water and maintain the conformation of biomolecules [47]. To demonstrate that the overexpression of *OsEF1A* and *LRK2* also possess similar traits, the relative water content, proline, malondialdehyde and chlorophyll contents were measured in wild-type and transgenic plants after 4 days of drought stress. The relative water content increased by approximately 6.64% in the overexpressing compared to wild-type strains, suggesting a greater water retention capacity in the transgenic plants. Meanwhile, the proline content increased by 5.91% in the overexpressing plants, further reflecting resistance to stress, since drought-resistant varieties tend to accumulate more proline [48,49]. In contrast, malondialdehyde, an organic substance produced by peroxidation of plant tissues or organ membranes under stress, decreased by approximately 19.46% in the overexpressing compared to wild-type plants, which as with proline, indicates stress tolerance [10,50]. Meanwhile, no significant difference in the chlorophyll content was observed between the overexpressing and wild-type plants.

Based on this study, overexpression of *OsEF1A* is thought to increase yield by increasing the effective number of tillers and number of grains per plant, as well as causing an increase in drought tolerance. *OsEF1A* is therefore considered a useful tool for crop improvements in arid lands. Further research is now required to determine the underlying mechanisms.

## 4. Materials and Methods

### 4.1. Plasmid Construction and Transformation

The full-length *OsEF1A* gene was amplified from the CDS of the rice cultivar Nipponbare using the specific primers 1300-OsEF-BamHI-F (ATGGGTAAGGAGAAGACGCACATCA) and 1300-OsEF-PstI-R (AGCGTAATCTGGAACATCGTATGGGTA). It was then inserted into the pCAMBIA 1300-2 × 35S vector and transferred into Nipponbare using Agrobacterium-mediated transformation to obtain *OsEF1A* overexpressing and antisense strains.

### 4.2. Subcellular Localization of OsEF1A

To determine the subcellular localization of *OsEF1A*, codon-free *OsEF1A* was amplified by PCR using the specific primers 1300-OsEF-F (ATGGGTAAGGAGAAGACGCACATCA) and 1300-OsEF-R (AGCGTAATCTGGAACATCGTATGGGTA). The amplification product was then purified and fused to the pCAMBIA1301-CFP vector using T4 ligase then the resulting *35S::CFP:OsEF1A* construct was introduced into onion epidermal cells using the gene gun method. Fluorescence signals were then observed using a laser scanning confocal microscope.

### 4.3. Promoter–GUS Analysis

To construct a *OsEF1A::GUS* vector, the promoter of *OsEF1A* was amplified from rice genomic DNA using the specific primers pBI121-pro-EF-HindIII-F (TTGAAGCTT AAGTCGGTTGAGATGCACCACG) and pBI121-pro-EF-BamHI-R (CCAGGATCC ATGAAAGGGCTTAACAGAAGTGGAAAC). The PCR product was then purified using T4 ligase and cloned into the pBI121 vector. The cloned vector was introduced into rice via Agrobacterium-mediated introduction. Root, node, basal stem, sheath samples and anther samples were then obtained at the seedling and flowering stages from the wild-type plants and *OsEF1A::GUS* plants for GUS staining.

### 4.4. Total RNA Extraction, cDNA Synthesis and Real-Time Quantitative PCR Analysis

Total RNA was extracted from the rice leaves using a RNA Extraction Kit (TIANGEN, Hefei, China) according to the manufacturer’s instructions then placed at −80 °C. The extracted RNA was then placed in PCR tubes, and cDNA was synthesized by reverse transcription using a Reverse Transcription Kit (CWBIO, Taizhou, China) according to the manufacturer’s instructions. RT-PCR and real-time fluorescence quantitative PCR were then performed using the diluted reaction products as templates.

### 4.5. Growing Conditions

Wild-type and transgenic seeds were soaked in water then placed in an artificial climate chamber at 37 °C for germination. After germination, seeds were then transferred to 96-well plates and incubated at 28 °C under a photoperiod of 16 h light/8 h dark. After 14 days of growth, they were subjected to drought stress induced by 20% PEG6000. Seedlings were then sampled at 0, 3, 6, 12 and 24 h, frozen in liquid nitrogen and placed in a freezer at −80 °C for total plant RNA extraction.

### 4.6. Drought Stress Treatment

A total of 50 wild-type and 50 transgenic seeds were soaked in distilled water and 20% PEG6000 solution then examined for germination 4 days after drought treatment. Rice seedlings grown under normal conditions were allowed to grow to the three-leaf stage (14 days) then treated with 20% PEG6000. The phenotypes of the rice seedlings were observed and photographed at different time points until the wild-type plants showed curled leaves, at which point culture was restored with water. After recovery, survival was calculated by counting the number of plants with green, healthy young leaves.

### 4.7. Measurements of Agronomic Traits

Agronomic traits were observed in the experimental field at Jinhua in Zhejiang province. At maturity, plant height and the effective number of tillers per plant were measured in the field in approximately 20 plants per plant line. Individual plants were also marked for seed collection. Collected seeds were counted for seed setting rate and the number of grains per plant, and full seeds were used to measure the 1000-grain weight.

### 4.8. Analysis of the Relative Water, Chlorophyll, Prolinecontent and Malondialdehyde Content

The relative water content, chlorophyll content, proline content and malondialdehyde content of two-week-old rice seedlings were measured on the fourth day of drought treatment, where the relative water content was determined by the method of Chen et al. [51]. Chlorophyll content, according to the method of Udawat et al. [52]; proline and malondialdehyde, according to the method of An et al. [53].

### 4.9. Data Analysis

Data were averaged under each experimental treatment and analyzed using SPSS 21 software with one- (ANOVA) or two-way analysis of variance (ANOVA) and S-N-K multiple comparison tests (differences were significant at *p* < 0.05). Excel 2019 software was used for all analyses.

## 5. Conclusions

In summary, the aim of this paper is to investigate the molecular characteristics and biological functions of the *OsEF1A* gene in rice. The results showed that *OsEF1A* was mainly localized in the cell membrane, cytoskeleton and nucleus. It was expressed in the root, root-stem union, the base of each stem node and anther. Overexpression of *OsEF1A* increased the effective number of tillers, the number of primary and secondary branching peduncles, single plant yield and grain length in rice, although the fruit set rate of rice was reduced. In addition, overexpression of *OsEF1A* could also improve the drought resistance of rice at the seedling stage to some extent, through changes in intracellular malondialdehyde, proline and relative water content in response to drought stress. This study will provide a promising genetic resource for rice breeding and pave the way for a new green revolution.

## Figures and Tables

**Figure 1 plants-12-02593-f001:**
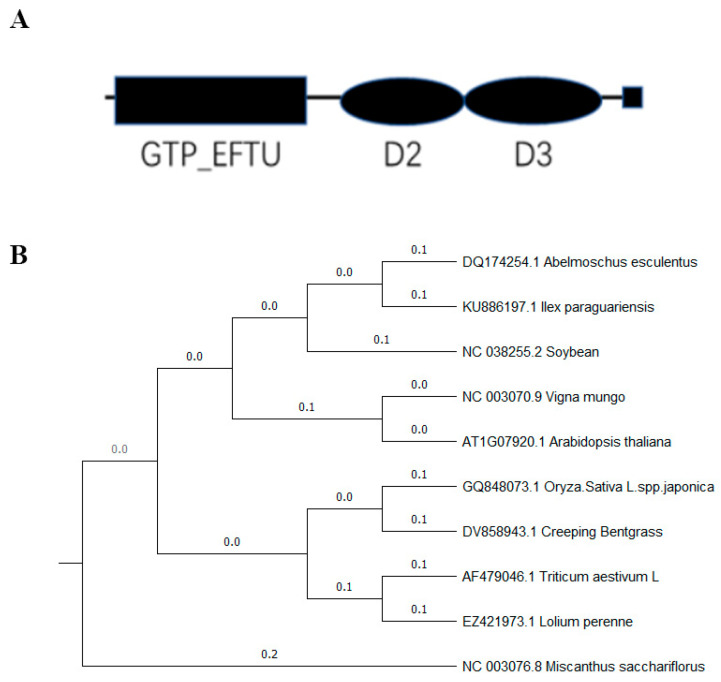
Sequence analysis of the *OsEF1A* gene. (**A**) Schematic diagram of the EF1A protein. (**B**) Phylogenetic analysis of the *EF1A* gene sequence with other homologous sequences.

**Figure 2 plants-12-02593-f002:**
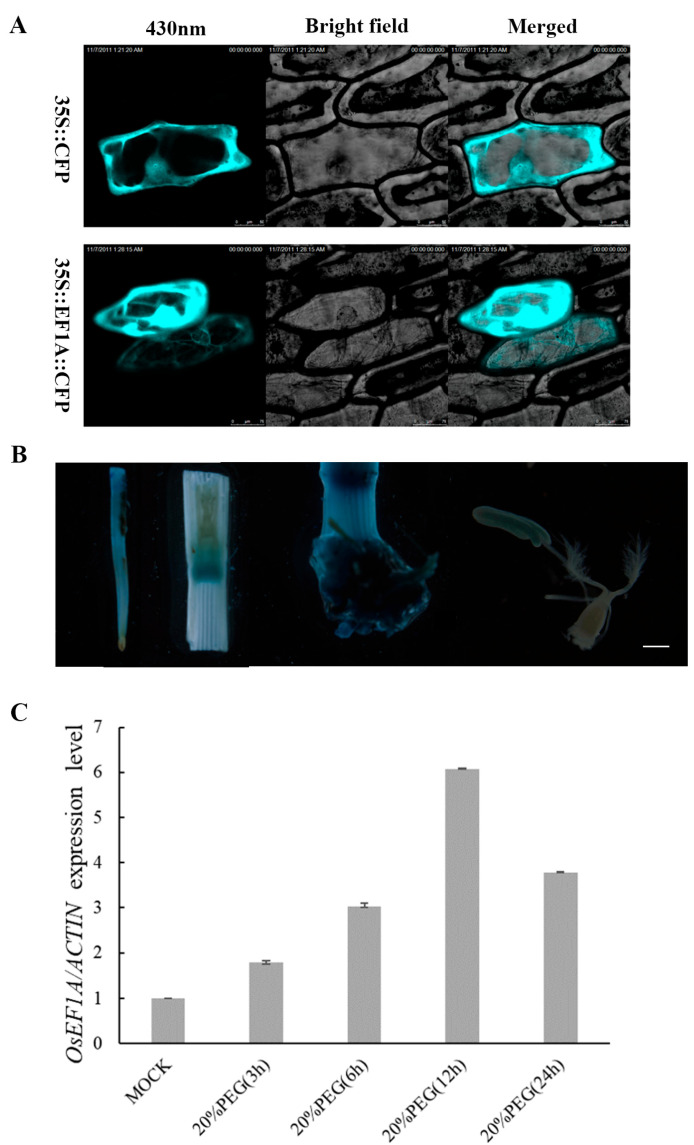
Subcellular localization and expression of *OsEF1A* in different rice tissues. (**A**) Subcellular localization of *OsEF1A*. (**B**) GUS staining of the roots, node, basal stem and anthers from GUS-positive plants (bar: 500 um). (**C**) Expression analysis of *OsEF1A* in wild-type rice seedlings under drought treatment using real-time PCR. Each column represents an average of three replicates and bars represent the margin of error (MOCK is 20%PEG 0h).

**Figure 3 plants-12-02593-f003:**
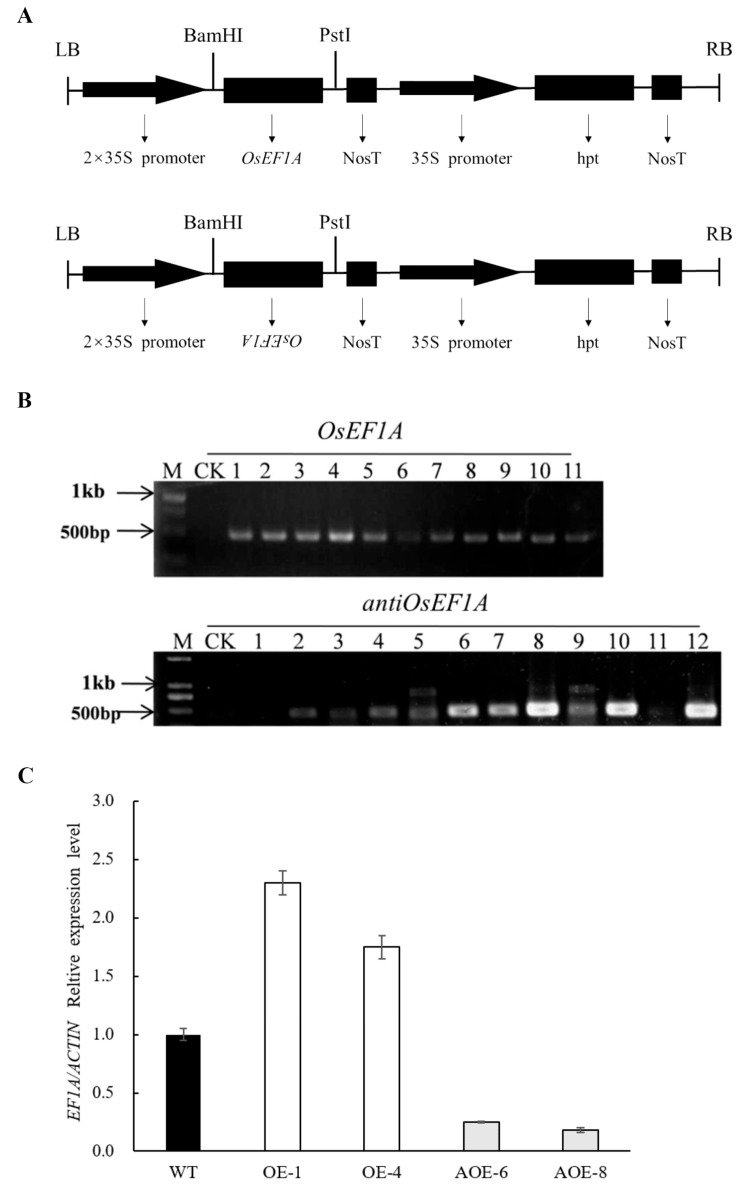
Expression of *OsEF1A* in transgenic rice (WT is wild type, OE-1 and 0E-4 are overexpression lines). (**A**) Schematic diagram of *OsEF1A* high- and low-expression vectors *pCAMBIA1300-2X35S::OsEF1A* and *pCAMBIA1300-2X35S::antiOsEF1A*, respectively. (**B**) Transgenic identification of *OsEF1A* and anti*OsEF1A*. (CK stands for wild type and the numbers represent different transgenic strains.) (**C**) Real-time PCR analysis of *OsEF1A* expression in transgenic plants.

**Figure 4 plants-12-02593-f004:**
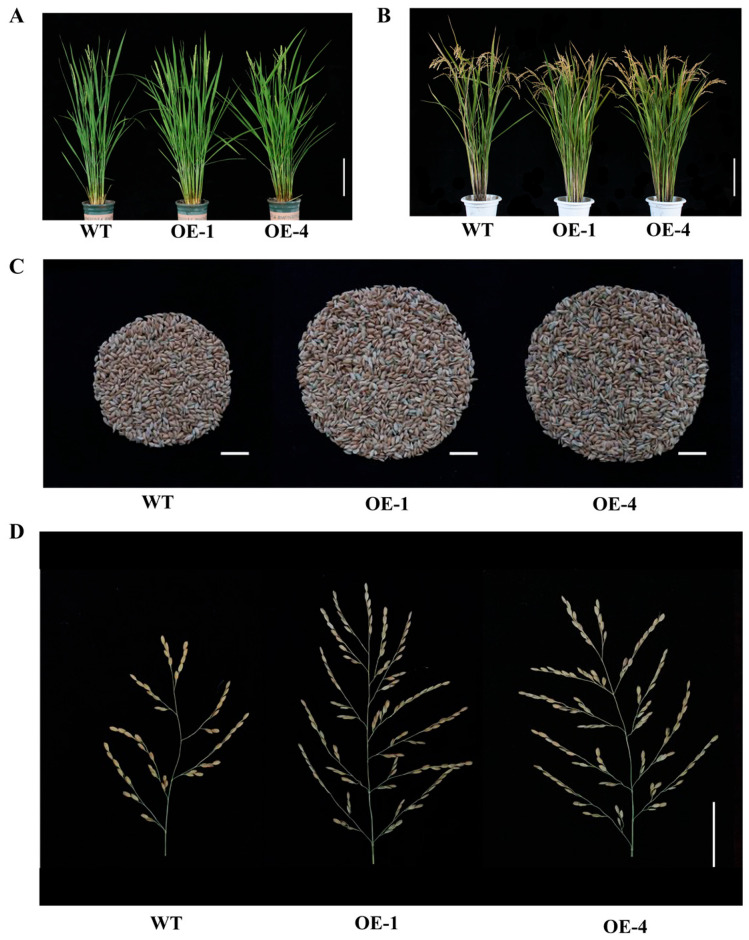
Phenotypes of wild-type and *OsEF1A*-overexpressing plants. (**A**) Comparison of wild-type (WT) and transgenic lines at the tillering stage (bar: 20 cm) and (**B**) at maturity stage (bar: 20 cm). (**C**) Comparison of the number of grains per plant (bar: 2 cm). (**D**) Primary and secondary branching peduncles (bar: 2 cm).

**Figure 5 plants-12-02593-f005:**
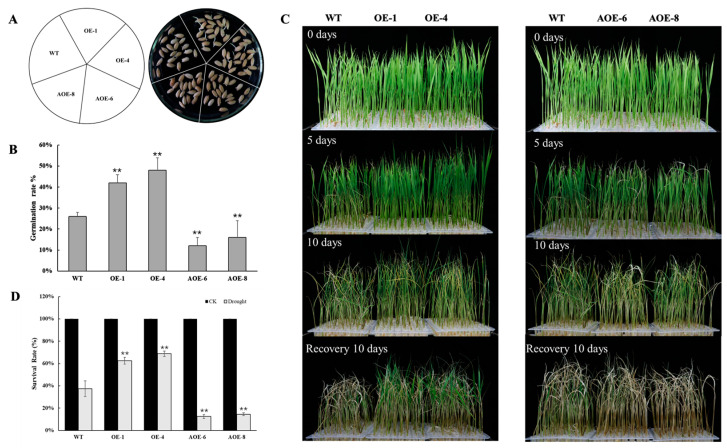
Performance of *OsEF1A* transgenic plants under drought stress. (**A**) Phenotypes of wild-type (WT) and transgenic lines following 20% PEG6000-induced drought treatment for 4 days. (**B**) Germination rates plants following 20%PEG6000 treatment. (**C**) Performance of the *OsEF1A* transgenic plants under drought stress. (**D**) Survival rates after recovery. ** is *p* < 0.01 based on the student’s *t*-test.

**Figure 6 plants-12-02593-f006:**
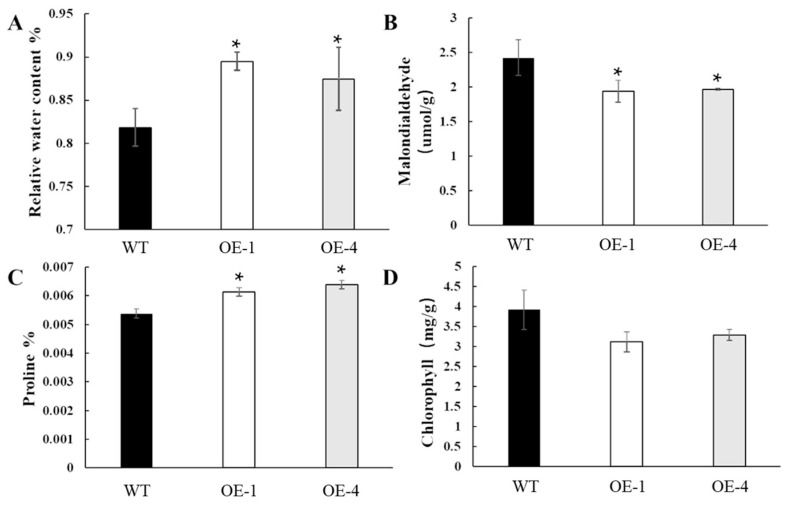
Physiological data on the fourth day of drought stress. (**A**) Relative water content, (**B**) malondialdehyde, (**C**) proline, (**D**) and chlorophyll contents of the wildtype and overexpressing plants. * *p* < 0.05 based on the student’s *t*-test.

**Table 1 plants-12-02593-t001:** Yield components in the wild-type (WT) and transgenic *OsEF1A*-overexpressing plants.

Plant Line	Tiller Number per Plant (cm)	Plant Height (cm)	1000-Grain Weight (g)	Seed Setting Rate (%)	Grain Number per Plant	Primary Branches	Secondary Branches
WT	16.23 ± 1.75	80.17 ± 2.94	25.68 ± 1.09	92.5 ± 4.7	1052.4 ± 123.7	7.83 ± 1.60	14.67 ± 2.42
OE-1	21.47 ± 2.25 **	72.42 ± 2.77 **	23.20 ± 1.02 **	85 ± 7.26 **	1371.4 ± 117.6 **	8.83 ± 1.47 *	17.52 ± 1.78 **
OE-4	20.32 ± 1.97 **	74.13 ± 2.15 **	23.07 ± 1.06 **	87 ± 6.54 **	1269.2 ± 101.9 **	8.22 ± 1.17 *	17.03 ± 1.26 **

*n* > 20, “*” is *p* < 0.05, “**” is *p* < 0.01, representing significant differences and based on the student’s *t*-test.

## Data Availability

We obtained the complete gene field and CDS sequence of *OsEF1A* from NCBI (https://www.ncbi.nlm.nih.gov/, accessed on 1 July 2023) and predicted the protein structure using the online software interpro (https://www.ebi.ac.uk/interpro/, accessed on 1 July 2023). All other data supporting this result are included in the article.

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
