# Peer review of "Eukaryotic Translation Elongation Factor OsEF1A Positively Regulates Drought Tolerance and Yield in Rice"

_plants, 2023, doi:10.3390/plants12142593_

Round 1
Reviewer 1 Report
Some concerns about this manuscript are listed below:
1) Please provide the reference of using 20% PEG6000 for drought stress treatment (line 90);
2) line 93, change to 7.4-fold and 3.3-fold
3) The legend of Fig. 2C, Mock should be 20% PEG6000, 0 h?
4) What is the 'n' values of the data with significant differences, like in Table 1 and Fig. 5, i.e. the number of biological replicates.
5) Why the 1000 grain weight decreased compared with the wild type, is there any explanation for it?
6) How about the phenotypes of OsEF1A-antisense under normal condition?
7) What is the drought stress treatment for the seed. germination in section 2.5?
8) Please add detail information in the Fig. 6 legend, like what does NIP and OE represent? and When did you measure the parameters?
9) Please combine the data of Table 1 and 2 as one table.
10) line 211, is there any reference or data of interaction between EF1A and LRK2?
Author Response
Dear Reviewer:
Thank you for your revision of the article on Eukaryotic translation elongation factor OsEF1A positively regulates drought tolerance and yield in rice, I have made specific changes and addressed the issues you raised. I have made specific changes and elaborated on your questions.
Question 1:Please provide the reference of using 20% PEG6000 for drought stress treatment (line 90);
Answer:A real-time analysis of the OsEF1A gene was performed after PEG drought treatment of wild-type rice (Haru, Japan), with the aim of understanding whether the expression of this gene would be elevated under drought. Here, only wild-type rice was used for the treatment.
Question 2:line 93, change to 7.4-fold and 3.3-fold
Answer:The usage of multiplier has been modified.
Question 3:The legend of Fig. 2C, Mock should be 20% PEG6000, 0 h?
Answer:Legend of Figure 2C, Mock is 20% of PEG6000 at 0h of treatment
Question 4:What is the 'n' values of the data with significant differences, like in Table 1 and Fig. 5, i.e. the number of biological replicates.
Answer :The number of biological replicates is probably more than 20 and the results have been counted for three years.
Question 5:Why the 1000 grain weight decreased compared with the wild type, is there any explanation for it?
Answer:Regarding the possibility of decreasing thousand grain weight, we guess that it is because the fertility of soil is limited while tillering increases, so the number of grains per plant increases but the thousand grain weight may decrease under the same environment, in addition to the possibility of decreasing pollen grain activity.Question 6:How about the phenotypes of OsEF1A-antisense under normal condition?
Answer:Under normal circumstances, the antisense phenotype is the same as the wild type, so it is not exhibited in the article.Question 7:What is the drought stress treatment for the seed. germination in section 2.5?
Answer:Drought treatments for seed germination were taken from 50 seeds each and also treated in 20% PEG 6000 solution so that resistance could be determined at an early stage.
Question 8:Please add detail information in the Fig. 6 legend, like what does NIP and OE represent? and When did you measure the parameters?
Answer:Modifications have been made, where NIP stands for wild type and OE for overexpression strain. All physiological data were repeated three times and more, and all measurements were made on the fourth day.Question 9:Please combine the data of Table 1 and 2 as one table.
Answer:Tables 1 and 2 have been combined for processing.
Question 10:line 211, is there any reference or data of interaction between EF1A and LRK2?
Answer:The interaction between EF1A and LRK2 was obtained in a previous article posted in our laboratory as reference 13, entitled: "Overexpression of the leucine-rich receptor-like kinase gene LRK2 increases drought tolerance and tiller number in rice"
Thank you in advance for reconsideration of our revised manuscript, and please let us know if there are any additional question, comments or concerns.
Best wishes.
Sincerely yours,
Qing Gu.
Reviewer 2 Report
The paper "Eukaryotic translation elongation factor OsEF1A positively regulates drought tolerance and yield in rice" aims to clarify the role of this elongation factor on rice regarding drought tolerance and yield.
The paper is technically sound and the methodology supports the results and the objective. However, the manuscript (MS) requires some revisions before its publication.
1. The authors should revise the species name through the whole MS. It is wrong to not use italics when writing scientific names such as O. sativa. For example: line 13 or 25. In line 25 Sativa should not contain any capital letter. Please revise the entire MS.
2. The keywords should avoid words that are already on title.
3. Authors should also check gene names that are not in italics throughout the MS.
4. In lines 34 and 35 the authors mention that a number of studies have addressed the selection of drought-tolerant genotypes but just a few are cited. In think that this revision should mention more studies.
5. The discussion need to be improved.
Author Response
Dear Reviewer:
Thank you for your revision of the article on Eukaryotic translation elongation factor OsEF1A positively regulates drought tolerance and yield in rice, I have made specific changes and addressed the issues you raised. I have made specific changes and elaborated on your questions.
Question 1:The authors should revise the species name through the whole MS. It is wrong to not use italics when writing scientific names such as O. sativa. For example: line 13 or 25. In line 25 Sativa should not contain any capital letter. Please revise the entire MS.
Answer:Errors that appear in the article have been corrected accordingly.
Question 2:The keywords should avoid words that are already on title.
Answer:The keywords section has been partially revised, but the whole article is probably based on the narrative of these keywords, I'm afraid there is no way to make major changes.
Question 3:Authors should also check gene names that are not in italics throughout the MS.
Answer:Errors that appear in the article have been corrected accordingly.
Question 4:In lines 34 and 35 the authors mention that a number of studies have addressed the selection of drought-tolerant genotypes but just a few are cited. In think that this revision should mention more studies.
Answer:The research on drought resistance genes actually consists of many studies, and some drought resistance genes have been added to this revision. This part of the revision may require some more articles in this field.
Question 5:The discussion need to be improved
Answer:I have finished adding words to the manuscript, and the additions are highlighted in red. Most of the modifications are in the discussion, including a detailed description of the study of drought genes and a listing of causes related to tillering, including and related hormonal changes. As well as detailing the range of physiological and biochemical reactions that occur in cells under drought stress.
Thank you in advance for reconsideration of our revised manuscript, and please let us know if there are any additional question, comments or concerns.
Best wishes.
Sincerely yours,